# Exclusive Breastfeeding for at Least Four Months Is Associated with a Lower Prevalence of Overweight and Obesity in Mothers and Their Children after 2–5 Years from Delivery

**DOI:** 10.3390/nu14173599

**Published:** 2022-08-31

**Authors:** Maria Mantzorou, Dimitrios Papandreou, Georgios K. Vasios, Eleni Pavlidou, Georgios Antasouras, Evmorfia Psara, Zainab Taha, Efthymios Poulios, Constantinos Giaginis

**Affiliations:** 1Department of Food Science and Nutrition, School of Environment, University of the Aegean, 81400 Myrina, Greece; 2Department of Health Sciences, College of Natural and Health Sciences, Zayed University, Abu Dhabi P.O. Box 144534, United Arab Emirates

**Keywords:** breastfeeding, overweight, obesity, body mass index, childhood obesity

## Abstract

Introduction: Obesity is a current public health concern. Higher body weight is influenced by genetic and environmental parameters, and their interplay and is associated with a greater risk for several chronic diseases. Breastfeeding has been suggested as a preventive measure against obesity, which can further reduce long-term negative health outcomes for both women and children. Aim: The aim of the present study was to evaluate the role of breastfeeding on maternal and childhood overweight and obesity. Materials and Methods: This is a cross-sectional study conducted on 2515 healthy mothers and their children, aged 2–5 years, enrolled from nine different Greek rural and urban regions. Validated, standardized questionnaires were administrated that included anthropometric indices, socio-demographic characteristics of mothers and children, as well as breastfeeding practices. Results: Overall, 68% of participated women exclusively breastfed their children for at least 4 months. Mothers that exclusively breastfed showed a significantly lower prevalence of overweight and obesity after 2–5 years from delivery (*p* < 0.0001). Children that had exclusively been breastfed showed a significantly lower prevalence of overweight and obesity at the age of 2–5 years (*p* < 0.0001). Using multivariate regression analysis, exclusive breastfeeding for at least 4 months was associated with a two-fold lower risk for maternal and childhood overweight and obesity after 2–5 years from delivery, independent from maternal age, educational and economic status, and smoking habits (*p* < 0.0001). Conclusion: Exclusive breastfeeding for at least 4 months had a positive effect on childhood overweight and obesity, also contributing beneficially to post-natal maternal weight control. The beneficial effects of breastfeeding should be communicated to future and new mothers, while supportive actions for all mothers to initiate and continue breastfeeding their offspring should be implemented.

## 1. Introduction

Obesity in adults and children is a current global public health concern that affects both the physical and the mental health of individuals [1,2], while it has a great cost to both the state and the individuals [3,4]. Current data reveal that worldwide, 2.1 billion people are overweight or obese, with obesity being the fifth leading cause of death [5]. Alarmingly enough, 41 million children under the age of 5 years were affected by overweight or obesity, and over 340 million children and adolescents aged 5–19 years were overweight or obese in 2016 [6]. According to World Health Organization (WHO), over 60% of children who are overweight before puberty continue to be overweight in early adulthood [7].

Obesity is multi-factorial and is affected by genetic and environmental parameters and their interplay [8,9,10,11,12], while in rare cases, obesity is attributed to genetic reasons only [13]. Actually, BMI-heritability ranges between 47–80% and is merely influenced by genetics [11], while the current obesogenic environment may amplify the predisposed genetic risk for obesity [14]. It has also been reported that children inherit about 40% of their parents’ BMI, with a greater proportion being passed on with greater offspring’s BMI [15]. Moreover, sleeping patterns, gut microbiota, and other factors have recently been identified as contributing factors in the development of obesity and the current obesity epidemic [16]. Hence, obesity prevention and management should be multi-dimensional [17], while the amelioration of the family lifestyle should be an integral part of the prevention and management strategies [12,18].

Prevention of obesity is crucial and can be initiated not only during childhood but also during pregnancy, as maternal obesity influences the offspring’s risk of developing obesity [19]. WHO recommends exclusive breastfeeding for “up to 6 months of age, with continued breastfeeding along with appropriate complementary foods up to 2 years of age or beyond” [20]. Breastfeeding has been associated with better long-term health status for both mothers and children [21]. Specifically, meta-analyses showed that breastfeeding may protect against child infections and malocclusion, increases intelligence, and possibly reduces overweight, asthma, and diabetes risk. In regard to mothers, breastfeeding has a protective effect in decreasing long-term negative health outcomes for women by reducing the risk for breast cancer as well as possibly ovarian cancer, cardiovascular diseases, hypertension, hyperlipidemia, and diabetes mellitus type II [21]. Breastfeeding also has a positive impact on maternal postnatal weight loss [22] and has been associated with lower risk of childhood obesity by 22% [23,24] and in a dose–response manner [25]. Yet, despite the benefits of breastfeeding for the child’s and mother’s health and body weight, mothers who are obese are less likely to breastfeed their children [26,27].

Several studies have indicated an inverse association between breastfeeding duration and the risk of maternal overweight and obesity [28,29,30]. In a Norwegian population, the probability of obesity in women under the age of 50 years who had never breastfed was more than 3-fold higher compared to those who had breastfed for at least 24 months [31]. Bobrow et al. also reported that in postmenopausal women, the mean BMI was 0.22 kg/m^2^ lower for each additional 6 months of breastfeeding [28]. In addition, Giesla et al. supported evidence that breastfeeding may have some beneficial, long-term effect on the risk of excessive weight and abdominal obesity in women [32]. Sharma et al. also found that women with obesity 6 years postpartum and who fully complied with the breastfeeding recommendations (i.e., exclusively breastfeeding for  ≥4 months and continuing breastfeeding for  ≥12 months) had a lower body mass than obese women who had never breastfed; however, they did not find such associations in women with normal weight or overweight [33]. On the other hand, some studies have not confirmed the association between the lactation duration and maternal weight [34,35].

As far as childhood overweight and obesity is concerned, Fallahzadeh et al. reported that breastfeeding for more than 24 months exerted a protective effect on children from becoming overweight [36]. An Australian survey including 2868 infants documented that children who were breastfed for less than 4 months had a significantly higher risk of childhood weight exceeding the 95th percentile of children of the same age and sex [37]. A Croatian study also indicated that breastfeeding for more than 6 months was a protective factor for overweight and obesity in children aged 6 to 11 years [38]. However, there are also studies that did not support significant associations between breastfeeding and prevention of maternal and childhood overweight and obesity, mainly when adjusting for confounding factors. For example, in a cohort study involving 5-year-old Swedish children, Huus et al. documented a non-association between exclusive breastfeeding and the risk of overweight and obesity [39]. Durmus et al. also reported no relationship in Dutch children at the age of 3 years [40]. These discrepancies among the different studies may be ascribed to the mothers’ and children’s age, parental country of birth, maternal age, smoking status, education level, and cultural differences.

Conclusively, even if there are many studies supporting the protective role of breastfeeding against maternal and childhood overweight and obesity, several other studies did not confirm these results, rendering the existing findings rather inconsistent and inconclusive and reinforcing the need for further investigation. In view of the above considerations, the aim of the present cross-sectional study was to evaluate the potential protective role of breastfeeding against maternal and childhood overweight and obesity in a representative Greek population sample, taking into account multiple confounding factors.

## 2. Materials and Methods

### 2.1. Subjects

In the present cross-sectional study, 2515 women and their children aged 2–5 years were enrolled from 9 different Greek rural and urban regions, namely Athens, Thessaloniki, Larisa, Patra, Alexandroupolis, Kalamata, Ioannina, Crete, and the North Aegean. Recruitment to the study was between the period May 2016 and September 2020. All participants’ information was confidential, and all participants were disease-free and informed about the purpose of the study and signed a consent form. In fact, mothers included in the study did not develop any disease during their pregnancy, such as gestational diabetes or pregnancy-induced hypertension, or during the postpartum period. Sample size calculation was based on the use of PS: Power and Sample Size calculator program, while the randomization was carried with the use of a sequence of random binary numbers (i.e., 001110110, in which 0 represented enrolment and 1 not enrolment to the study) [41]. The study was approved by the Ethics Committee of the University of the Aegean (ethics approval code: no 12/14.5.2016) and was in compliance with the World Health Organization (52nd WMA General Assembly, Edinburgh, Scotland, 2000). The exclusion criterion was any disease for enrolled women and their children. All children included in the present study had no health conditions that needed special care, such as breathing trouble, heart problems, infections, or birth defects, in order to be admitted to neonatal intensive care unit (NICU) after delivery.

### 2.2. Study Design

Validated, standardized questionnaires were used to assess the sociodemographic characteristics and anthropometric measures of the mothers and their children. These questionnaires assessed semi-quantitatively the sociodemographic characteristics and anthropometric measures of the participants [42,43]. Anthropometry included the sex, weight, height, and age of the children and mothers. Mothers self-reported their weight shortly before, during, and right after pregnancy as well as their children’s weight and height at birth. Maternal and childhood anthropometric measures (body weight and height) were measured by a trained nutritionist as per protocol 2–5 years after delivery [42,43]. Weight was measured using the same electronic scale, and height was measured using a portable stadiometer. Two international datasets are used to define overweight and obesity in pre-school children: the International Obesity Task Force (IOTF) reference and the WHO standard [6,44].

Considering mothers, their smoking habit, educational level, and the economic status were recorded. In fact, educational level was scaled according to the sum of years of education. Within the Greek education system, 15 years of education equates to high school graduation. Financial status was classified according to the annual income as: 0 ≤ EUR 5000, 1 ≤ EUR 10,000, 2 ≤ EUR 15,000, 3 ≤ EUR 20,000, 4 ≤ EUR 25,000, and 5 ≥ EUR 30,000 (Exchange rate: EUR 1.0 = USD 0.99). The information of breastfeeding was recorded by mothers’ self-reporting. In fact, mothers were asked whether they breastfed their children, whether they were exclusively breastfeeding, and the duration of breastfeeding.

Clarifying instructions were given to the participants by registered dietitians and physicians regarding the completion of questionnaires, while a detailed presentation of the questions to facilitate accurate answers was performed.

## 3. Statistical Analysis

Statistical analysis was performed by Student’s *t*-test and one-way ANOVA for continuous variables found to follow the normal distribution by the use of Kolmogorov–Smirnov test. Chi-square test was used for categorical variables. Mann–Whitney non-parametric test was used for non-normally distributed continuous variables between two groups, while Kruskal–Wallis non-parametric test was applied for non-normally distributed variables between three or more groups. The normally distributed quantitative variables are presented as mean value ± standard deviation (SD) and the qualitative variables as absolute or relative frequencies. Multivariate logistic regression analysis was performed to assess the influence of exclusive breastfeeding on maternal and childhood overweight and obesity after adjustment for potential confounders, e.g., age, educational and economic status, and smoking habits. Differences were considered significant at *p* < 0.05 and 95% confidence interval. The statistical analysis of the survey data was performed by SPSS 21.0 program (Statistical Package for Social Sciences, Chicago, IL, USA).

## 4. Results

### 4.1. Sociodemographic Characteristics and Anthropometric Measures of the Participant Mothers and Their Children

The present study enrolled 2515 mothers and their children, aged 2–5 years. The mean age of the mothers was 37.51 ± 4.88 years. Concerning their nationality, 95.8% of the mothers were Greek, and the remaining 4.2% were of other ethnicities (Albanian, Russian, Ukrainian, Bulgarian, Pakistani, African). As far as the educational level of the mothers is concerned, the mean years of education were 15.09 ± 2.23 years (range: 6–17 years of education). With regard to the economic level, 45% of the participant women reported lower financial status than the medium one for Greek citizens, and 55% had higher financial status than the medium one. Moreover, 25.7% of the women were smokers both pre-pregnancy and after 2–5 years from delivery.

The mean body mass index (BMI) of the mothers before pregnancy was 22.7 ± 3.7 kg/m^2^, while at the time of the study (2–5 years postpartum), the BMI was significantly higher at 23.8 ± 4.4 kg/m^2^ (*p* < 0.0001). More to the point, pre-pregnancy, 18.1% of the women were overweight, and 5.2% were obese, according to their BMI. Mothers that did not breastfeed showed higher pre-pregnancy BMI than those who breastfed after delivery (Figure 1A, 22.9 ± 3.6 vs. 22.5 ±3.2 kg/cm^2^, *p* = 0.0167). At the time of study (2–5 years postpartum), 22.5% of the women were overweight, and 12.7% were obese, according to their BMI.

As far as children are concerned, their mean age was 4.04 ± 1.06 years (range: 2–5 years), and their mean body weight at birth was 3182 ± 515 g (range: 1320–5000 g). At the age of 2–5 years, 74.5% of the children had normal BMI, while 16.2% were overweight, and 7.7% were obese.

### 4.2. Exclusive Breastfeeding for at Least 4 Months and Its Association with Sociodemographic Characteristics of the Participant Mothers

Most mothers (68%) did breastfeed exclusively for at least 4 months (mean duration: 4.6 ± 1.8 months), and 32% did not exclusively breastfeed for at least 4 months or did not breastfeed at all. Women who chose to breastfeed exclusively for at least 4 months were younger than those who did not breastfeed exclusively for at least 4 months or did not breastfeed at all (Figure 1B, Table 1, 37.1 ± 4.6 vs. 38.1 ± 5.1 years, *p* < 0.0001). Moreover, those who chose to breastfeed exclusively had higher educational levels than those who did not breastfeed exclusively or at all (Table 1, 15.3 ± 2.1 vs. 14.7 ± 2.3 years, *p* < 0.0001). Furthermore, the economic status of women who did breastfeed exclusively for at least 4 months was significantly higher than that of women who did not breastfeed exclusively for at least 4 months or did not breastfeed at all (Table 1, *p* = 0.0322). Similarly, women who chose to breastfeed exclusively for at least 4 months were smokers at lower rates than women who did not breastfeed for at least 4 months or did not breastfeed at all (Table 1, *p* = 0.0195).

### 4.3. Exclusive Breastfeeding for at Least 4 Months and Its Association with Maternal and Childhood Anthropometric Measures

Women who breastfed exclusively for at least 4 months had slightly lower pre-pregnancy BMI than those who did not breastfeed exclusively for at least 4 months or did not breastfeed at all (Figure 1B, 22.5 ± 3.7 kg/m^2^ vs. 22.9 ± 3.7 kg/m^2^, *p* = 0.0167). Women that breastfed exclusively for at least 4 months had lower BMI 2–5 years post-delivery than those who did not breastfeed exclusively for at least 4 months or did not breastfeed at all (Figure 1C, Table 1, 23.6 ± 4.3 kg/m^2^ vs. 24.1 ± 4.5 kg/m^2^, *p* = 0.0008). Concerning BMI classes, among the mothers that exclusively breastfed for at least 4 months, 20.0% of them were overweight, and 6.4% were obese. Among the mothers who did not exclusively breastfeed for at least 4 months or did not breastfeed at all, a significantly higher prevalence of overweight (22.5%) and obesity (12.7%) was recorded compared to mothers that exclusively breastfed for at least 4 months (Table 1, *p* < 0.0001).

Exclusive breastfeeding rates were progressively higher with decreasing children’s BMI rates. More to the point, children who had exclusively been breastfed for at least 4 months had lower BMI rates at the age of 2–5 years than those who did not breastfeed exclusively for at least 4 months or did not breastfeed at all (Figure 1D, Table 1, 15.7 ± 1.9 kg/m^2^ vs. 16.4 ± 2.3 kg/m^2^, *p* < 0.0001). Concerning BMI classes, among children that had exclusively been breastfed for at least 4 months, 16.2% of them were overweight, and 7.7% were obese. Among children that had not been exclusively breastfed for at least 4 months or had not been breastfed at all, a significantly higher prevalence of overweight (20.2%) and obesity (10.7%) was recorded compared to children that had exclusively been breastfed for at least 4 months (Table 1, *p* < 0.0001).

Younger mothers, of other ethnicities, with high educational and economic level, and no smokers significantly and more frequently breastfed exclusively for at least 4 months in unadjusted analysis (Table 1, *p* < 0.0001, *p* = 0.0001, *p* < 0.0001, *p* = 0.0032 and *p* = 0.0195, respectively).

### 4.4. Multivariate Regression Analysis Assessing the Influence of Exclusive Breastfeeding on Maternal and Childhood Overweight and Obesity

In the multivariate logistic regression analysis, mothers who exclusively breastfed for at least 4 months showed more than two-fold lower risk for presenting overweight or obesity 2–5 years post-delivery regardless of confounding factors, such as maternal age, educational and economic status, and smoking habit (Table 2, *p* < 0.0001). Younger mothers had a 52% higher likelihood to exclusively breastfeed than older mothers (Table 3, *p* = 0.0023); however, this association was attenuated by the adjustment of the confounding factors. Mothers with high educational level had a 77% higher probability to exclusively breastfeed; however, this association was also attenuated by the adjustment of the confounding factors. Moreover, children who had exclusively been breastfed for at least 4 months showed more than two-fold lower risk for being overweight or obese at the age of 2–5 years old regardless of confounding factors, such as maternal age, BMI, educational and economic status, and smoking habit (Table 3, *p* < 0.0001). Younger mothers with high education level showed a 61% and a 83% higher probability for exclusively breastfeeding in the adjusted regression model for children. Maternal nationality, economic status, and smoking habits did not remain significant in both adjusted regression models ( Table 2; Table 3, *p* > 0.05).

## 5. Discussion

In the present study, 22.5% of the women were classified as overweight and 12.7% as obese, according to their BMI. The prevalence is surprisingly lower than expected. According to the WHO [7], over 50% of both men and women in the WHO European Region were overweight, and almost 23% of women and 20% of men were affected by obesity. Furthermore, according to the findings of a previous Greek study [45], 29.3% of the women were overweight, and 25.6% were obese.

Considering children, we found that 7.7% were obese, and 16.2% were overweight. According to the WHO, the prevalence of childhood overweight and obesity in Europe varies among different European countries, from 11% to 33%, with the highest rates in southern countries, such as Greece, where the highest rate was observed [44]. In Greece, the latest data come from the EYZHN study [46], with 336,014 children aged 4–17 years, which found that at 4 years, 5.8% of boys and 5.7% of girls were obese, and 10.4% of boys and 15.5% of girls were overweight. At age 5, 8.5% of boys and 8.3% of girls were obese, and 13.2% of boys and 15.8% of girls were overweight.

Directly after birth, 56–98% of infants in 11 European countries receive human milk, while at 6 months, 38–71% and 13–39% of infants were breastfed or exclusively breastfed, respectively [47]. The WHO European Region has the lowest rates of exclusive breastfeeding at 6 months in the world, at approximately 25% [47]. Exclusive breastfeeding in Greece at 6 months is the lowest in the WHO European Region at 0.7% [44]. Surprisingly, in the present study, the majority of the mothers did breastfeed their children for 8.9 ± 7.9 months, and 68% of women exclusively breastfed for 4.6 ± 1.8 months. This finding may be due to the fact that mothers who do choose to breastfeed are more likely to be better-informed about breastfeeding and more willing to participate in research studies, and especially in the last few years.

Currently, there are several studies supporting the protecting role of breastfeeding against maternal and childhood overweight and obesity postpartum [28,29,30,36,37,38]; however, other studies did not confirm any significant association [34,35,39,40], rendering overall the existing findings rather inconsistent and inconclusive and reinforcing the need for further investigation.

In this aspect, the results of our study are largely consistent with those studies that indicated a positive effect of breastfeeding on reducing the risk of maternal overweight and obesity. More to the point, we found that women who breastfed their offspring had lower BMI after 2–5 years after delivery than women who did not breastfeed. Moreover, those who exclusively breastfed for at least 4 months had lower BMI than those who did non-exclusively breastfed for at least 4 months or did not breastfeed at all. Postpartum weight loss due to breastfeeding has been well-documented [22,48], and it is a motivation for women to breastfeed their children [49]. Moreover, our findings suggest that mothers with higher BMI were breastfeeding their children less. This finding is in accordance with several reviews [26,50] that reported that higher maternal BMI is negatively associated with breastfeeding initiation and duration. In support of this view, in the recent study by Ramji et al. [27] conducted on 12,422 women, progressively decreasing rates of breastfeeding were highlighted with increasing BMI class. In fact, 71.1% of normal-weight mothers breastfed their children, while just 54.2% and 42.3% of morbidly obese and extremely obese women breastfed their children, respectively. The reasons of not breastfeeding may be cultural, psychological, or physiological [26,51]. In line with our findings, several previous studies have also indicated a protective effect of breastfeeding against the risk of maternal obesity [28,29,30]. In a Norwegian population, the prevalence of obesity in women under the age of 50 years who had never breastfed was more than 3-fold higher than those who had breastfed for at least 24 months [31]. In addition, Bobrow et al. reported that in postmenopausal women, the mean BMI was 0.22 kg/m^2^ lower for each additional 6 months of breastfeeding [28].

The benefit of a healthier weight due to breastfeeding, and especially exclusive breastfeeding, also extends to children. We found that children aged 2–5 years who were breastfed had lower BMI than those who did not breastfeed, and those who were exclusively breastfed for at least 4 months had lower BMI at 2–5 years than those who did not breastfeed or did not breastfeed exclusively. In support of this view, breastfeeding and exclusive breastfeeding has been suggested as a protective factor against childhood overweight and obesity [24,52,53]. In fact, a meta-analysis of 25 studies with 226,508 participants from 12 countries noted the association with a 22% reduced risk of obesity in children, while categorical analysis of 17 studies showed that there was a dose–response effect between breastfeeding duration and risk of childhood obesity [24]. Moreover, an earlier study also showed a dose-dependent association between breastfeeding duration and lower childhood overweight risk [25]. In addition, breastfeeding at 1 month and for longer than 6 months reduced the risk of obesity by 36–52% during childhood and early adolescence, as shown in a longitudinal study with 1234 U.S. children [54]. Furthermore, a recent retrospective cohort study documented that exclusive breastfeeding for 6 months is a protective factor for childhood obesity in high-birthweight children [55]. Additionally, Yeung et al. [56] found that exclusive breastfeeding for 4 months was associated with less increase in weight-for-length percentiles in the first year of life. In accordance with our findings, Tambalis et al. [57], in a Greek sample, found that exclusive breastfeeding for more than 6 months versus never was associated with lower risk of childhood obesity by 30% and adolescent obesity by 38%, indicating the long-term positive effect of exclusive breastfeeding. In a more recent meta-analysis including 26 prospective cohort studies, breastfeeding was inversely associated with the risk of early obesity in children aged 2–6 years [58]. In this meta-analysis, a dose–response effect between duration of breastfeeding and reduced risk of early childhood obesity was noted [58]. Thus, the authors strongly recommended clinical nurses’ guidance and advice to prolong the duration of breastfeeding and promote exclusive breastfeeding with the aim at preventing the development of later childhood obesity [58].

Although the present study presents data that support the importance of the breastfeeding on weight control of mothers and their children, we have to underline several limitations of our study. Specifically, this is a retrospective study, and hence, associations but not causation can be derived from the data. In this aspect, the results of our study should be further confirmed by large-scale prospective studies. The study sample is limited and includes women who live in certain Greek regions. Thus, larger, long-term epidemiological studies with samples from other areas of the country, e.g., urban, rural, and island regions, are essential in order for more reliable conclusions regarding the Greek population to be drawn. The recall bias is also a limitation of the study. The recall bias may be prevented due to the reliability of the questionnaire and the guidance provided to women during the completion of the questionnaire by qualified dietitians and physicians. In addition, the present study has several strengths since detailed clarifying instructions were given to the participants by registered dietitians and physicians regarding the completion of questionnaires, while a detailed presentation of the questions to facilitate accurate answers was performed. Furthermore, all participants, both mothers and children, were disease-free. All children had no health conditions at birth that needed special care in NICUs. Moreover, mothers included in the study did not develop any disease during their pregnancy, such as gestational diabetes or pregnancy-induced hypertension, which may confound the effect of breastfeeding on maternal and childhood overweight and obesity 2–5 years after birth.

## 6. Conclusions

In conclusion, our cross-sectional study, in accordance with earlier studies, found that exclusive breastfeeding for at least 4 months has a protective role both for postpartum maternal weight loss and against childhood overweight and obesity. Taking into consideration the plethora of benefits attributed to breastfeeding and the fact that mothers with higher body weight are less likely to breastfeed, it is imperative to inform them concerning the benefits of exclusive breastfeeding for themselves and their children for at least 4 months and support all mothers to initiate and continue breastfeeding their offspring.

## Figures and Tables

**Figure 1 nutrients-14-03599-f001:**
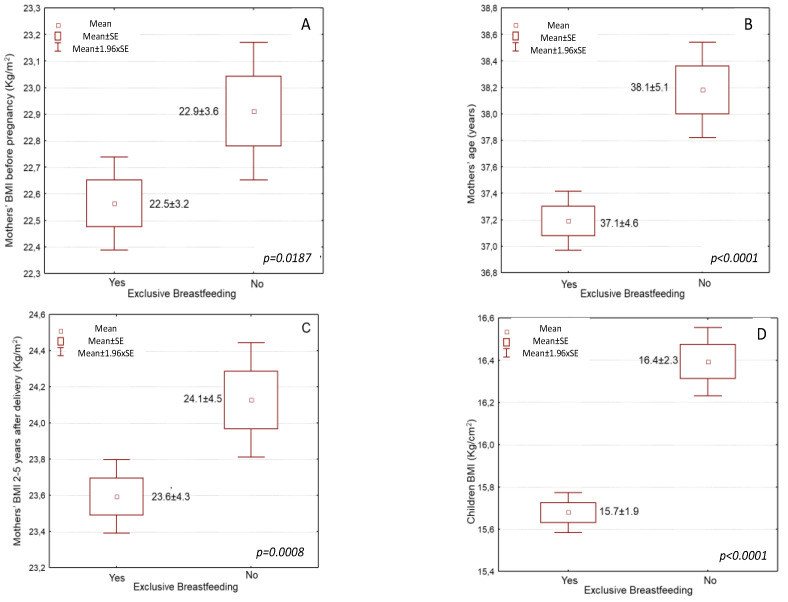
Associations of exclusive breastfeeding for at least four months with (**A**) mothers’ BMI before pregnancy, (**B**) mothers’ age at the time of study, (**C**) mothers’ BMI after 2–5 years from delivery, and (**D**) children’s BMI at the time of study.

**Table 1 nutrients-14-03599-t001:** Associations of exclusive breastfeeding for at least 4 months with sociodemographic and anthropometric characteristics for mothers 2–5 years after delivery and their children.

Sociodemographic and Anthropometric Characteristics	Exclusive Breastfeeding		
	No	Yes	*p*-Value
Mothers’ age (years)	38.1 ± 5.1	37.1 ± 4.6	*p* < 0.0001
Mothers’ nationality (% Greek)	48.7	47.1	*p* = 0.0001
Mothers’ BMI (kg/m^2^)	24.1 ± 4.5	23.6 ± 4.3	*p* = 0.0008
Mothers’ BMI (% overweight or obese)	35.2	33.1	*p* < 0.0001
Mothers’ education (years)	14.7 ± 2.3	15.3 ± 2.1	*p* < 0.0001
Mothers’ economic status (scale 0–5)	3.58 ± 0.9	3.64 ± 1.1	*p* = 0.0322
Mothers’ smoking habits (% smokers)	51.30	48.70	*p* = 0.0195
Children age (years)	4.07 ± 1.06	4.03 ± 1.04	*p* = 0.2613
Children gender (%male)	49.8	49.9	*p* = 0.5347
Children BMI (kg/m^2^)	16.4 ± 2.3	15.7 ± 1.9	*p* < 0.0001
Children BMI (% overweight or obese)	30.9	27.2	*p* < 0.0001

**Table 2 nutrients-14-03599-t002:** Multivariate logistic regression analysis for assessing the influence of exclusive breastfeeding on maternal overweight and obesity after adjustment for potential confounders.

Sociodemographic and Anthropometric Characteristics	Exclusive Breastfeeding	
	HR * (95% CI **)	*p*-Value
Age (below/over mean value)	1.52 (0.881–2.319)	*p* = 0.0023
Nationality (Greek/other nationality)	0.88 (0.262–1.632)	*p* = 0.0537
BMI (normal/overweight or obese)	2.14 (1.871–2.427)	*p* < 0.0001
Education (below/over mean value)	1.77 (0.944–2.558)	*p* = 0.0045
Economic status (below/over mean value)	1.64 (0.375–2.983)	*p* = 0.3781
Smoking habits (No/Yes)	2.87 (1.591–3.294)	*p* = 0.1941

* HR, Hazard ratio; ** CI, confidence interval.

**Table 3 nutrients-14-03599-t003:** Multivariate logistic regression analysis for assessing the influence of exclusive breastfeeding on childhood overweight and obesity after adjustment for potential confounders.

Sociodemographic and Anthropometric Characteristics	Exclusive Breastfeeding	
	HR * (95% CI **)	*p*-Value
Mothers’ age (below/over mean value)	1.61 (0.74–2.419)	*p* = 0.0082
Children gender (male/female)	1.08 (0.125–2.376)	*p* = 0.4891
Mothers’ nationality (Greek/other nationality)	0.72 (0.102–2.391)	*p* = 0.1349
Children BMI (normal/overweight or obese)	2.07 (1.704–2.398)	*p* < 0.0001
Mothers’ education (below/over mean value)	1.83 (0.827–2.901)	*p* = 0.0321
Mothers’ economic status (below/over mean value)	1.42 (0.176–3.479)	*p* = 0.6287
Mothers’ smoking habits (No/Yes)	2.54 (0.876–4.176)	*p* = 0.5279

* HR, hazard Ratio; ** CI, confidence interval.

## Data Availability

Not applicable.

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
