# Peer review of "Exclusive Breastfeeding for at Least Four Months Is Associated with a Lower Prevalence of Overweight and Obesity in Mothers and Their Children after 2–5 Years from Delivery"

_nutrients, 2022, doi:10.3390/nu14173599_

Round 1
Reviewer 1 Report
What is the other ethnicity of the 4.2%?
Should interpret what 15 years of education equates to within the Greek education system
Also, need to contextualize the income (meaning interpret for us what that means in dollars or Greek currency); otherwise, the reader won’t know
The Discussion section simply restates the results, but this is the place where you need to talk about your results in relation to other research that is out there and what they have found
It would be helpful to state the research questions at the end of the introduction section, so that people have an idea of what the study is looking to investigate.
A better use of incorporating other research in the introduction and discussion section is needed.
Author Response
Reviewer 1
We would like the thank the reviewer for its valuable comments and suggestions which were taken into full consideration in order for improving our manuscript.
- What is the other ethnicity of the 4.2%?
Response: We now reported the other ethnicities.
- Should interpret what 15 years of education equates to within the Greek education system
Response: We explained what 15 years of education equates to within the Greek education system
- Also, need to contextualize the income (meaning interpret for us what that means in dollars or Greek currency); otherwise, the reader won’t know
Response: We added the exchange rate between euros and dollars.
- The Discussion section simply restates the results, but this is the place where you need to talk about your results in relation to other research that is out there and what they have found
Response: We added several statements in the discussion including studies that we previously did not report them, and we compared them with our findings. Similarly, we enriched the introduction with additional relevant studies related with the basic aim of our study.
- It would be helpful to state the research questions at the end of the introduction section, so that people have an idea of what the study is looking to investigate.
Response: We added several sentences in the introduction to state more clear what our study is looking to investigate. We emphasized the controversies existed in the previous studies which reinforce the need for further studies.
- A better use of incorporating other research in the introduction and discussion section is needed.
Response: We incorporated additional research studies in both the introduction and discussion in order to state more clear the need of our study and to compare the existing knowledge with our results.
Reviewer 2 Report
The study examined the role of breastfeeding on maternal and childhood overweight and obesity through a large sample size investigation in eight cities of Greece. The results showed exclusive breastfeeding for at least 4 months was associated with a two-fold lower risk for maternal and childhood overweight and obesity after 2-5 years from delivery. The study findings strengthened the beneficial effects of breastfeeding to both mothers and children, and suggested the significance of supportive actions for all mothers to initiate and continue breastfeeding. However, the writing of the manuscript should be improved. There are some key issues to be classified by authors.
1. The significance of preventing maternal obesity should be strengthened in the background, not only for reducing the risk of childhood obesity but also decreasing long term negative health outcomes for women.
2. What is the study design? Is it a cross-sectional or prospective study design? Could authors clearly state the study design in methodology section. It seems that some variables such as birth weight were drawn from the medical recordings. Some other variables were measured at 2-5 years after birth.
3. The sample size calculation should be described in details rather than saying by use of the sample size calculation program.
4. How did the study obtain the information of breastfeeding? This key information is absent in the manuscript. It was described to use validated questionnaires to measure anthropometric measures of the participants. What were those questions? How to ensure these questionnaires were valid?
5. How maternal or women’s weight and height were measured? These are very important indicators of this study. It should be introduced very clearly.
6. Are there other important factors should be included in the analysis, such as health status of newborn, the admission of NICU etc.
7. What’s the added value of this study?
8. I did not see the detailed tables of the results.
Author Response
Reviewer 2
We would like the thank reviewer for its valuable comments and suggestions which were taken into full consideration in order for improving our manuscript.
- The significance of preventing maternal obesity should be strengthened in the background, not only for reducing the risk of childhood obesity but also decreasing long term negative health outcomes for women.
Response: We emphasized both in abstract and introduction the significance of preventing maternal obesity, reporting also the decreasing risk for long term negative health outcomes for women through breastfeeding.
- What is the study design? Is it a cross-sectional or prospective study design? Could authors clearly state the study design in methodology section. It seems that some variables such as birth weight were drawn from the medical recordings. Some other variables were measured at 2-5 years after birth.
Response: We stated in both abstract, introduction and study design sections that this is a cross-sectional study. We also specified which anthropometric variables were self-reported by the mothers and which anthropometric variables were measured at 2-5 years after birth by experts. We also specified the criteria provided by two international datasets to define overweight and obesity in pre-school children.
- The sample size calculation should be described in details rather than saying by use of the sample size calculation program.
Response: We added a new reference cited in study design concerning the sample size calculation performed in our study.
- How did the study obtain the information of breastfeeding? This key information is absent in the manuscript. It was described to use validated questionnaires to measure anthropometric measures of the participants. What were those questions? How to ensure these questionnaires were valid?
Response: We further reported in the study design that the information of breastfeeding was recorded by mothers’ self-reporting, and we discussed this fact in the limitations of our study. We also added the two citations which ensure the valid use of the questionnaires used: 1) World Health Organization, Physical status: the use and interpretation of anthropometry. Report of a WHO Expert Committee. World Health Organ Tech Rep Ser, 1995. 854: p. 1-452. 2) Prevention, C.f.D.C.a., National Health and Nutrition Examination Survey (NHANES) Anthropometry Procedures Manual. 2007.
- How maternal or women’s weight and height were measured? These are very important indicators of this study. It should be introduced very clearly.
Response: Maternal and childhood anthropometric measures (body weight and height) were measured by trained nutritionists and physicians as per protocol 2-5 years after delivery according to the cited references: 1) World Health Organization, Physical status: the use and interpretation of anthropometry. Report of a WHO Expert Committee. World Health Organ Tech Rep Ser, 1995. 854: p. 1-452. 2) Prevention, C.f.D.C.a., National Health and Nutrition Examination Survey (NHANES) Anthropometry Procedures Manual. 2007. We have already speculated that weight was measured using the same electronic scale, and height was measured using a portable stadiometer.
- Are there other important factors should be included in the analysis, such as health status of newborn, the admission of NICU etc.
Response: We added a statement reporting that all children included in the present study had non health condition that needed special care such as breathing trouble, heart problems, infections, or birth defects in order to be admitted to neonatal intensive care unit (NICU). We further stated in the study design that mothers included in the study did not develop any disease during their pregnancy such as gestational diabetes or pregnancy induced hypertension or during postpartum period.
- What’s the added value of this study?
Response: We added the strengths of our study below its limitation in order to emphasize its added value.
- I did not see the detailed tables of the results.
Response: We apologize for that. We had included our tables as separate files. We now included them inside the manuscript.